# Small Fiber Neuropathy in Sarcoidosis

**Natalia Gavrilova** [1,2], **Anna Starshinova** [2,*], **Yulia Zinchenko** [1,2], **Dmitry Kudlay** [3,4], **Valeria Shapkina** [5], **Anna Malkova** [2], **Ekaterina Belyaeva** [1,6], **Maria Pavlova** [1], **Piotr Yablonskiy** [1,2] and **Yehuda Shoenfeld** [2,7,8]

1   St. Petersburg Scientific Research Institute of Phthisiopulmonology, 191036 Saint-Petersburg, Russia; fromrussiawithlove_nb@mail.ru (N.G.); ulia-zinchenko@yandex.ru (Y.Z.); ekaterina_83@bk.ru (E.B.); mvpavlova2011@mail.ru (M.P.); piotr_yablonskii@mail.ru (P.Y.)
2   Laboratory of the Mosaic of Autoimmunity, Saint-Petersburg State University, 199034 Saint-Petersburg, Russia; anya.malkova.95@mail.ru (A.M.); yehuda.shoenfeld@sheba.health.gov.il (Y.S.)
3   Department of Pharmacology, Institute for Pharmacy, Sechenov University, 119571 Moscow, Russia; D624254@gmail.com
4   Laboratory of Personalized Medicine and Molecular Immunology, Institute of Immunology, 115552 Moscow, Russia
5   National Medical Research Center for Oncology, 197758 Saint-Petersburg, Russia; valshapkina@gmail.com
6   City Tuberculosis Hospital №2, 194214 Saint-Petersburg, Russia
7   Zabludowicz Center for Autoimmune Diseases, Sheba Medical Center, Tel HaShomer, Ramat Gan 52621, Israel
8   Sackler Faculty of Medicine, Tel-Aviv University, Tel Aviv 6997801, Israel
*   Correspondence: starshinova_777@mail.ru

**Abstract:** Sarcoidosis (SC) is a granulomatous disease of an unknown origin. The most common SC-related neurological complication is a small fiber neuropathy (SFN) that is often considered to be the result of chronic inflammation and remains significantly understudied. This study aimed to identify the clinical and histological correlates of small fiber neuropathy in sarcoidosis patients. The study was performed in 2018–2019 yy and included 50 patients with pulmonary sarcoidosis ($n = 25$) and healthy subjects ($n = 25$). For the clinical verification of the SFN, the "Small Fiber Neuropathy Screening List" (SFN-SL) was used. A punch biopsy of the skin was performed followed by enzyme immunoassay analysis with PGP 9.5 antibodies. Up to 60% of the sarcoidosis patients reported the presence of at least one complaint, and it was possible that these complaints were associated with SFN. The most frequent complaints included dysfunctions of the cardiovascular and musculoskeletal systems and the gastrointestinal tract. A negative, statistically significant correlation between the intraepidermal nerve fiber density (IEND) and SFN-SL score was revealed. In patients with pulmonary sarcoidosis, small fiber neuropathy might develop as a result of systemic immune-mediated inflammation. The most common symptoms of this complication were dysautonomia and mild sensory dysfunction.

**Keywords:** sarcoidosis; autoimmune inflammation; polyneuropathy; small fiber neuropathy; autoimmunity



## 1. Introduction

In patients with sarcoidosis, the development of systemic inflammation and internal organ dysfunction are observed, which significantly reduces quality of life and worsens patient prognosis. One of the most common complications is considered to be small fiber neuropathy (SFN), a condition that remains significantly understudied [1–4]. Quite often, patients with lung sarcoidosis complain of a number of non-specific symptoms, such as weakness, sleep disorders, etc., which can significantly affect the patient's quality of life, even in the absence of significant deviations from the pulmonary system.

The development of SFN is considered to be the result of cytokine-mediated inflammation, which is typical for various autoimmune diseases, including sarcoidosis [5–8]. Small nerve fiber damage is also observed in systemic lupus erythematosus, Sjogren's

syndrome, and fibromyalgia [9,10]. Considering the significant role of genetic predisposition and the possible provocative role of exogenous triggers in the development of this complication, SFN in patients with sarcoidosis can be considered as a part of the autoimmune/inflammatory syndrome that is induced by adjuvants (ASIA) [11]. Several proven cases SFN have also been observed in patients suffering from the bacterial inflammation, e.g., Lyme disease and leprosy [1–4].

The prevalence of sarcoidosis varies throughout the world. In Japan, there is 1 case per 100,000 people, while in Scandinavian countries, the prevalence of sarcoidosis is as much as 63 cases per 100,000 people. In the Russian Federation, sarcoidosis is described to have a prevalence of 22 to 47 cases per 100,000 people, depending on the region [12–15]. The prevalence of SFN may also vary because it not only presents with neuropathic pains and paresthesias, but also with various symptoms of autonomic dysfunction, which may not be recognized as a neurologic complication [16–20].

Currently, there are no generally accepted criteria for the diagnosis of SFN. The presence of a small nerve fiber dysfunction in a patient is mainly based on clinical criteria, such as a neurological examination and validated scales (the small fiber neuropathy screening list, for example) [19,21–23]. In addition, electroneuromyography can be performed to ensure that limited damage is induced in large nerve fibers. The "gold standard" for diagnostics is the immunofluorescence or immunohistochemistry of a skin biopsy that allows the density of the intraepidermal nerve fibers to be calculated. This technique requires special training and equipment and is time consuming [24–30]. A preliminary clinical diagnosis of SFN in patients with sarcoidosis is important due to the low awareness of healthcare practitioners about this complication as well as to determine a quick assessment for neuropathic signs and to increase the performance of skin biopsies in diagnosing his condition.

## 2. Materials and Methods

A prospective comparative study was performed in 2018–2019 yy at the St. Petersburg Research Institute of Phthisiopulmonology and at the St. Petersburg City Public Health Institution "City Multi-disciplinary Hospital No. 2". The study was approved by the independent ethics committee of the St. Petersburg Research Institute of Phthisiopulmonology (extracted from protocol No. 46.1 of 04/20/2018). All of the study participants signed an informed consent.

The study included 50 patients with lung sarcoidosis and healthy subjects (average age was $38.4 \pm 7.2$ y.o.). The first group consisted of patients with lung sarcoidosis ($n = 25$, average age $33.4 \pm 8.5$ y.o.) and healthy subjects ($n = 25$, average age $43.2 \pm 11.7$ y.o.). There were no statistically significant differences in gender and age in the patient groups. The inclusion criteria were age between 18 and 65 y.o. and signing the informed consent to participate in the study. The patients from group 1 were diagnosed with stage I-II sarcoidosis. The diagnosis of pulmonary sarcoidosis was performed according to the criteria of the American Thoracic Society (ATS), the European Respiratory Society (ERS), and the World Association of Sarcoidosis and Other Granulomatous Diseases (WASOG).

The exclusion criteria were systemic glucocorticoid therapy, the presence of Löfgren's syndrome a chronic disease course (for patients with sarcoidosis), the presence of other infectious diseases (HIV, hepatitis C), a history of cancer that was treated with chemotherapy, the presence of other diseases (diabetes, hypothyroidism, renal failure, vitamin deficiency or overdose), the use of medications (metronidazole, ni-trofurantoin, linezolid, flecainide, statins) as well as a medical history of autoimmune diseases.

According to the design of the study, the patients with sarcoidosis underwent a routine examination using X-ray, morphological, bacteriological, and molecular genetic methods. A neurological examination assessing superficial and deep pain sensitivity, muscle strength, and muscle-tonic reflexes was performed. For the clinical verification of the SFN, the validated questionnaire for the detection of small fiber neuropathy was used (Small Fiber Neuropathy Screening List, SFN-SL, ([19],). The questionnaire consists of two parts and has

21 questions (Table S1). Questions are ranked on a scale from 0 to 4 points each and evaluate both the frequency at which symptoms develop and their intensity. A moderate likelihood of neuropathy is established when the diagnostic threshold is reached at 22 points, and a high diagnostic threshold is determined at 48 points. With scores of less than 11 (points), the sensitivity is 100%, and the specificity is 31.0%. When scores of more than 18 are achieved, then the sensitivity is 19.0%, and the specificity is 100%.

A total of 23 patients with lung sarcoidosis agreed to undergo a skin punch biopsy (10 cm proximal to the external malleolus), which was then fixed in n Zamboni solution, and the performance of the enzyme immunoassay with the primary PGP 9.5 antibodies (Abcam) and the secondary AlexaFluor goat-anti rabbit antibodies 488 (Abcam) was evaluated. The results were compared to the normal values that were obtained in the worldwide normative reference study [23].

Statistical analysis was performed using the Statistica 8.0 software package (Statsoft, USA). The distribution of patients into groups was carried out based on the presence of a verified diagnoses of pulmonary sarcoidosis or based on the absence of verified somatic diseases (for healthy subjects).

All of the data that were collected during the study were analyzed using descriptive statistical methods. Normally distributed data are presented in the form of the "mean ± standard deviation" formula. Data that were not normally distributed are presented in the form of the "median (interquartile range 25—75 quartiles)", where the confidence interval was also calculated. Normality testing was performed using the Shapiro–Wilk test.

All of the data were analyzed using parametric and nonparametric statistical methods and the Chi-square test, and statistical comparison methods for two (Mann–Whitney U-test) and three (Kruskal–Wallis test) groups were performed; correlations were determined using the Spearman coefficient. Differences or association rates were considered to be statistically significant at a $p < 0.05$.

### 3. Results

The average score of the SFN screening list scale (SFN-SL) in patients with sarcoidosis was 2.0 (0.0;7.0), and in healthy volunteers, it was 0 (0;0) points (Table 1).

**Table 1.** The results of the "Small Fiber Neuropathy Screening list" (SFN-SL) testing in patients with pulmonary sarcoidosis and healthy volunteers.

| Group | % (*n/N*) | SFN-SL Scale Points, Median (IQR) | CI 95% |
|---|---|---|---|
| Sarcoidosis (*n* = 25) | 60.0 * <br> 15/25 | 2.0 (0.0;7.0) | 1.836–2.748 |
| Healthy volunteers (*n* = 25) | 8.0 <br> 2/25 | 0 (0;0) | 0.00–0.00 |

* $p < 0.0001$.

Statistically significant differences were found in the results of patients with sarcoidosis and in the results of healthy subjects ($p < 0.0001$). Thus, in the first group, a higher average SFN-SL score was observed than it was in the second group, while the results of Groups 1 and 2 exceeded those of healthy individuals.

The main clinical symptoms of SFN were pain, changes in body temperature, impaired motility of the gastrointestinal tract and urinary system, and heart palpitations (Table 2).

In patients with sarcoidosis, the most frequent clinical symptoms included impaired cardiovascular regulation (palpitations, dizziness), pain in the chest or in the extremities, and blurred vision. It is important to note that in 76% of patients with sarcoidosis, the clinical symptoms had a severity of no more than 1 point on the SFN-SL scale and, in most cases, these symptoms did not bother patients or lead to a decrease in quality of life (Table 3).

**Table 2.** Clinical symptoms of small fiber neuropathy in patients with pulmonary sarcoidosis and healthy volunteers.

| | Sarcoidosis, (*n* = 25), *n* (%) | CI 95% | Healthy Volunteers, (*n* = 25), *n* (%) | CI 95% | $p^2$ |
|---|---|---|---|---|---|
| Cardiovascular disorders | 9 (36.0) | 34.5–38.0 | 0 | 0 | 0.005 |
| Gastrointestinal disorders | 3 (12.0) | 11.8–12.3 | 1 (0.04) | 0.8–1.9 | 0.084 |
| Urinary disorders | 0 | 0 | 0 | 0 | - |
| Musculoskeletal disorders | 4 (16.0) | 15.5–16.7 | 1 (0.04) | 0.7–1.8 | 0.020 |
| Skin and mucous membranes disorders | 7 (28.0) | 27.6–29.0 | 0 | 0 | - |
| Ophtalmologic disorders | 5 (20.0) | 19.4–20.9 | 0 | 0 | 0.005 |

**Table 3.** Clinical manifestations of small fiber neuropathy in patients with sarcoidosis and healthy volunteers.

| Clinical Manifestation of SFN | Sarcoidosis, (*n* = 25), *n* (%) | | | | Healthy Volunteers, (*n* = 25), *n* (%) | | | |
|---|---|---|---|---|---|---|---|---|
| | Points | | | | Points | | | |
| | 1 | 2 | 3 | 4 | 1 | 2 | 3 | 4 |
| Cardiovascular disorders | 4 (16.0) | 2 (8.0) | 3 (12.0) | 0 | 0 | 0 | 0 | 0 |
| Gastrointestinal disorders | 2 (8.0) | 0 | 1 (4.0) | 0 | 0 | 1 (0.08) | 0 | 0 |
| Urinary disorders | 0 | 0 | 0 | 0 | 0 | 0 | 0 | 0 |
| Muscular spasms | 2 (8.0) | 2 (8.0) | 0 | 0 | 1 (0.08) | 0 | 0 | 0 |
| Pain syndrome | 3 (12.0) | 1 (4.0) | 0 | 2 (8.0) | 0 | 0 | 0 | 0 |
| Temperature dysfunction | 1 (4.0) | 0 | 0 | 1 (4.0) | 0 | 0 | 0 | 0 |
| Skin discoloration | 3 (12.0) | 0 | 0 | 1 (4.0) | 0 | 0 | 0 | 0 |
| Paresthesias | 1 (4.0) | 0 | 1 (4.0) | 0 | 0 | 0 | 0 | 0 |
| Blurred vision | 2 (8.0) | 2 (8.0) | 0 | 1 (4.0) | 0 | 0 | 0 | 0 |
| Dry mucous membranes, change in skin moisture | 1 (4.0) | 0 | 0 | 0 | 0 | 0 | 0 | 0 |
| Results | 19 (76.0) | 7 (28.0) | 5 (20.0) | 5 (20.0) | 1 (1.4) | 1 (0.08) | 0 | 0 |
| CI 95% | 75.6–77.4 | 27.6–29.0 | 19.5–21.1 | 18.3–20.8 | 0.9–1.9 | 0.9–1.9 | 0 | 0 |

A total of 13 biopsy specimens were obtained from the patients with sarcoidosis, where the intraepidermal density (IEFD) of the small nerve fibers was calculated (Figures 1 and 2).

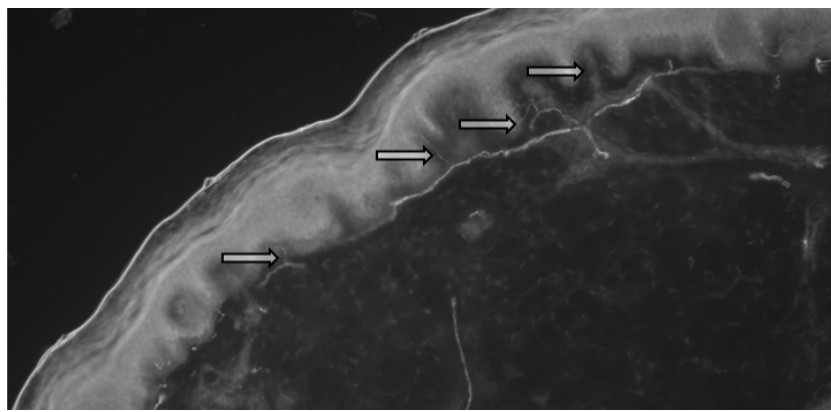

**Figure 1.** Skin biopsy of patient with sarcoidosis (enzyme immunoassay, male, 34 years). Arrows indicate small fibers in the epidermis of the skin.

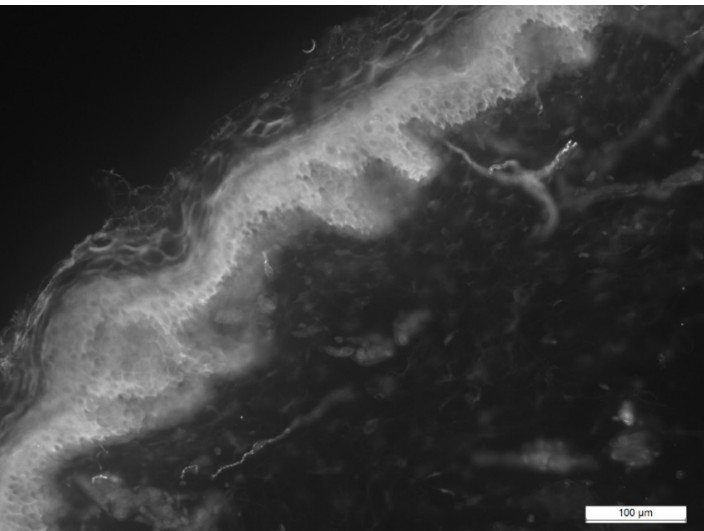

**Figure 2.** Skin biopsy of patient with sarcoidosis (enzyme immunoassay, male, 34 years).

A total of 13 of the biopsies were obtained from patients with sarcoidosis (Table 4). The calculations of the intraepidermal density of the small fibers were performed in accordance with the values obtained from the worldwide normative reference study [23].

**Table 4.** Intraepidermal nerve fiber density (IEND) in patients with sarcoidosis.

| Group | IEND in 1 mm, Median (Q1; Q3) | Normal Values—Median | Normal Values—0.05 Quantile |
|---|---|---|---|
| Pulmonary sarcoidosis ($n$ = 13) | 7.68 (7.02;8.34) | 12.4 | 7.1 |

Thus, in all of the examined patients, the number of small nerve fibers was within normal limits but below average values. A negative, statistically significant correlation was revealed between the IEND results and in the SFN-SL scores of patients with sarcoidosis (Spearman's nonparametric rank correlation coefficient, r = −0.3508, $p$ = 0.0102). Additionally, a negative, statistically significant correlation between the IEND and SFN-SL score was described (Spearman coefficient, r = −0.3508, $p$ = 0.0102).

## 4. Discussion

In our study, complaints that are typical for the small fiber neuropathy were described in 60% of the patients with sarcoidosis. Small fiber neuropathy can manifest with a wide range of symptoms, including autonomic and sensory dysfunction. The most common clinical manifestations that were observed in our study in the patients with sarcoidosis were impaired cardiovascular regulation (36% of cases), e.g., the development of palpitations and dizziness. In 32% of cases, patients noted pain in the chest or limbs, which was often accompanied by allodynia, a subjective perception of tactile touch as pain. In some cases, this can result in sleep disturbances due to the pain sensations that are caused by bed linen touching the skin. Another symptom that is often noted by patients with sarcoidosis is blurred vision, which is described in 20% of cases. A physician needs to clarify whether these visual impairments are transient or permanent in nature in order to determine a differential diagnosis of ophthalmic pathology. In the neuropathy of small fibers, blurred vision is transient, arising while overworking or during physical exertion. Gastrointestinal tract disorders are less commonly reported with patients with sarcoidosis. In 12% of the cases, the patients complained of impaired intestinal motility, with the development of both diarrhea and constipation, which occurred simultaneously with the onset of sarcoidosis. This also includes subjective complaints of swallowing dysfunction, which is associated with both impaired muscle innervation and with the progression of mouth dryness.

While a negative, statistically significant correlation between the IEND and SFN-SL scores was described in both groups (Spearman coefficient, r = −0.3508, *p* = 0.0102, and r = −0.7382, *p* = 0.0064), a decrease in the density of the small nerve fibers in patients with pulmonary sarcoidosis was more prominent.

A major limitation of this study was that conventional nerve conduction studies and autonomic function tests were not performed. The small sample size also could also be considered as a limitation.

Thus, the neuropathy of small fibers seems to be a widespread pathology that results in the development of multiple organ dysfunction. Disorders in small fiber neuropathy, along with the typical complaints of patients with lung sarcoidosis, such as cough and shortness of breath, can significantly contribute to a decrease in the quality of life. At the same time, when focusing on the main complaints as well as on instrumental and laboratory deviations, clinicians often do not attach the necessary importance to the manifestations of SFN. It is necessary to identify the main causes that are responsible for a patient's quality of life deteriorating in order to determine a treatment strategy and to improve the prognosis of the patient's disease [31,32].

Given the low awareness of both medical specialists and patients about the development of this complication and the difficulties that are involved in its diagnosis, further study into this issue is required.

In patients with pulmonary sarcoidosis, small fiber neuropathy may develop as a result of systemic immune-mediated inflammation. The validated questionnaires and histologic verification of the diagnosis help to establish the severity of the neuropathy of small fibers to determine the prognosis and to plan the treatment strategy.

**Supplementary Materials:** The following are available online at https://www.mdpi.com/article/10.3390/pathophysiology28040035/s1, Table S1: Small fiber neuropathy screening list questionnaire.

**Author Contributions:** All authors contributed to the study conception and design. Material preparation, data collection, and analysis were performed by N.G., A.M. and V.S. Patient selection and supervision were performed by Y.Z., M.P., E.B. and D.K. General supervision was carried out by A.S. and P.Y. General research concept, guidance, and article review was performed by Y.S. All authors have read and agreed to the published version of the manuscript.

**Funding:** This work was supported by a grant from the Government of the Russian Federation (contract No. 14W03.31.0009 dated 13 February 2017) on the allocation of the grant for the state support of scientific researchers conducted under the guidance of leading scientists.

**Institutional Review Board Statement:** All human studies have been approved by the independent ethics committee of the St. Petersburg Research Institute of Phthisiopulmonology (extract from protocol No. 46.1 of 20 April 2018) and have been performed in accordance with the ethical standards laid down in the 1964 Declaration of Helsinki and its later amendments.

**Informed Consent Statement:** Informed consent was obtained from all subjects involved in the study.

**Data Availability Statement:** The data presented in this study are available on request from the corresponding author. The data are not publicly available due to their containing information that could compromise the privacy of research participants.

**Conflicts of Interest:** The authors declare no conflict of interest.

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
