# Peer review of "Small Fiber Neuropathy in Sarcoidosis"

_pathophysiology, doi:10.3390/pathophysiology28040035_

Round 1

Reviewer 1 Report

The idea of the paper is very interesting, but the material and methods shall be improved. Authors should explain as enrollment is organized (are subjects with new diagnosis consectuveily enrolled?) and whether subjects with tubercolosis are also enrolled. Are controls matched for gender and age? Why did you select subjects with sarcoidosis stage I-II? Are you sure that all enrolled subjects with sarcoidosis have SFN? For instance, cardiovascular symtoms can be due to heart direct involvement of sarcoidosis- Is the mthod of bioosy validated? If only some subjects with sarcoidosis have SFN, you cannot include all groups in a table to establish percentages

I do not understand the lines 77-91. This paragraph includes a comment that I also approve

The Materials and Methods should be described with sufficient details to allow others to replicate and build on the published results. Please note that the publication of this manuscript implicates that you must make all materials, data, computer code, and protocols associated with the publication available to readers. New methods and protocols should be described in detail while well-established methods can be briefly described and appropriately cited.

The following phrase is a nonsense in this paper "Research manuscripts reporting large datasets that are deposited in a publicly available database should specify where the data have been deposited and provide the relevant  accession numbers. If the accession numbers have not yet been obtained at the time of submission, please state that they will be provided during review. They must be provided prior to publication.

Author Response

Dear colleague,

Thank you very much for your corrections! 

The study included patients with a first diagnosed pulmonary sarcoidosis stage I-II (the diagnosis was established according to the criteria of ATS/ERS/WASOG).

The patients with tuberculosis were not enrolled, corrected.

 Controls are matched for gender and age.

The design of our study implied the inclusion of patients with a newly diagnosed pulmonary sarcoidosis. During the examination, patients with stage I-II were identified and were included in the study.

We conducted this study to identify SFN among patients with sarcoidosis, but did not include patients with already established SFN.

All patients underwent ultrasound of the heart and ECG, at the time of examination there were no signs of sarcoid heart lesion. Perhaps such suspicions will appear in the follow-up period.

Is the mthod of bioosy validated? – Yes, this is a standardized method for SFN diagnostics, which is the «gold standard».

If only some subjects with sarcoidosis have SFN, you cannot include all groups in a table to establish percentages - we have corrected

This paragraph includes a comment that I also approve.

The questionnaires used for the diagnosis of SFN, as well as the protocols of biopsy and histological examination, are validated. References to the materials are indicated in the publication (23-25, 27, 29, 31, 32).

The following phrase is a nonsense in this paper "Research manuscripts reporting large datasets that are deposited in a publicly available database should specify where the data have been deposited and provide the relevant  accession numbers. If the accession numbers have not yet been obtained at the time of submission, please state that they will be provided during review. They must be provided prior to publication.- corrected

Reviewer 2 Report

1- Grammar and English :

Page 1 line 18 add: "Here we tasked " before "to identify the clinical"

Page 1 line 20 move "years" Before 2018-2019

Etc.. In fact there are many grammatical errors too many for me to address.

The instruction on the Methods has not been deleted from the manuscript. Please have the article corrected by a  native speaker. I will focus on the scientific issues.

2-In the method section please explain why the chronic sarcoid patients where excluded and why only stage I-II patients enrolled?

3-Please explain why tuberculosis is mentioned in the abstract and results? Are you suggesting concomitant disease?

4- If possible place a comparison  slide of a patient without SFN and show the differences.

5- Again not sure of the data presented for Tuberculosis? please explain how is this relating to the problem at hand?

6-Perhaps in introduction indicate some of the common symptoms of SFN in general, and in the results or discussion compare with the Sarcoid SFN

7-The healthy controls seem to have been selected with attempt to match the cases? Please provide a comparison of the typical characteristics of patients in the Sarcoidosis vs. Control. If the difference is significant (specially with regards to age and sex and comorbidities) these differences have to be discussed.

Author Response

1- Grammar and English :

Page 1 line 18 add: "Here we tasked " before "to identify the clinical" - Added

Page 1 line 20 move "years" Before 2018-2019 – Corrected

Etc.. In fact there are many grammatical errors too many for me to address.

The instruction on the Methods has not been deleted from the manuscript - corrected

Please have the article corrected by a native speaker. I will focus on the scientific issues. The language has been checked by a specialist and improved.

2-In the method section please explain why the chronic sarcoid patients where excluded and why only stage I-II patients enrolled? The design of our study implied the inclusion of patients with a newly diagnosed pulmonary sarcoidosis. During the examination, patients with stage I-II were identified and were included in the study.

3-Please explain why tuberculosis is mentioned in the abstract and results? Are you suggesting concomitant disease? No, the patients with tuberculosis were not enrolled in the study, corrected.

4- If possible place a comparison  slide of a patient without SFN and show the differences. We presented the results according to the research and analysis protocol, which did not include this calculation. We will plan this analysis for future studies

5- Again not sure of the data presented for Tuberculosis? please explain how is this relating to the problem at hand? The patients with tuberculosis were not enrolled in the study, corrected.

6-Perhaps in introduction indicate some of the common symptoms of SFN in general, and in the results or discussion compare with the Sarcoid SFN

The structure we have presented reflects the course of our thoughts and it seems important to us to maintain exactly this order

7-The healthy controls seem to have been selected with attempt to match the cases? Please provide a comparison of the typical characteristics of patients in the Sarcoidosis vs. Control. If the difference is significant (specially with regards to age and sex and comorbidities) these differences have to be discussed. The control group is represented by healthy individuals without chronic diseases, matched by gender and age to the main group.
